

# Epidemiological dynamics of an urban Dengue 4 outbreak in São Paulo, Brazil

Christian Julián Villabona-Arenas[1], Jessica Luana de Oliveira[1,2], Carla de Sousa-Capra[3], Karime Balarini[4], Celso Ricardo Theoto Pereira da Fonseca[4] and Paolo Marinho de Andrade Zanotto[1]

[1] Laboratory of Molecular Evolution and Bioinformatics, Department of Microbiology, Biomedical Sciences Institute, University of São Paulo, São Paulo, Brazil
[2] Department of Biomedicine, University of Mogi das Cruzes, Mogi das Cruzes, São Paulo, Brazil
[3] Office of Epidemiological Surveillance, Department of Health of Guarujá, Guarujá, São Paulo, Brazil
[4] Clinical Laboratory Analysis Center, ITAPEMA, Guarujá, Brazil

## ABSTRACT

**Background:** Dengue studies at the urban scale are scarce and required for guiding control efforts. In Brazil, the burden of dengue is high and challenges city public health administrations with limited resources. Here we studied the dynamics of a dengue epidemic in a single city. **Methods:** Serum samples from dengue suspected cases were collected and tested, from December 2012 and July 2013 in Guarujá, Brazil. We use incidence series analysis to provide a detailed view of the reproduction number dynamics and a Bayesian analysis to infer the spread of the serotype using geographic and temporal data. **Results:** We obtained nucleotide sequences from 354 envelope genes and georeferenced 286 samples during the course of the outbreak. Serotype 4 was responsible for the epidemic. We identified at least two major lineages that overlapped in distribution. We observed high reproduction numbers and high cladogenesis prior to the escalation of clinical case notifications. Three densely populated non-adjacent neighborhoods played a pivotal role during the onset and/or course of the epidemic. **Discussion:** Our findings point to high dengue virus transmission with a substantial proportion of unapparent cases that led to a late recognition of an outbreak. Usually source reductions initiatives tend to be insufficient once an epidemic has been established. Nevertheless, health authorities in Guarujá prioritized vector control on specific places with clusters of georeferenced viremic patients, which appear to have diminished the epidemic impact.

## INTRODUCTION

The dengue viruses exist as four antigenically distinct serotypes named DENV-1, DENV-2, DENV-3 and DENV-4. Dengue fever (DF) is a disease caused by any of the DENV (*Chen & Vasilakis, 2011*). There is an estimate of 390 million (95% credible interval 284–528) dengue infections worldwide per year, of which 96 million (67–136) show any level of clinical or subclinical severity (*Bhatt et al., 2013*).

Corresponding author
Paolo Marinho de Andrade Zanotto, pzanotto@usp.br

Dengue is endemic in Brazil. This means that the disease occurs every year, usually during the wet season when *Aedes* mosquitoes' population sizes are high and the rainfall is optimal for breeding. People provide the mosquitoes not only with blood meals but also water-holding containers where the mosquitoes lay their eggs. In addition, this country is at periodic risk for epidemic dengue (*i.e.*, when large numbers of people become infected during a short period), which requires a coincidence of large numbers of vector mosquitoes and large numbers of people with no immunity to one or more of the four serotypes (*CDC, 2015*).

Dengue virus serotype 4 (DENV-4) reemerged in the northern Brazil in 2010, 28 years after it was last detected in the country in 1982, and has been responsible for several outbreaks since then (*Nunes et al., 2012*). In 2013, 1,468,873 dengue cases were reported countrywide, including 6,969 severe cases and 545 deaths. These numbers entail a challenge for public health authorities, which in a timely manner need to allocate resources and trained personnel to try diminishing the health impact of the disease. Programs to control populations of mosquitoes strain public resources, especially in resource-limited settings (*Shepard et al., 2011*; *Stahl et al., 2013*). In this context, understanding epidemic spread in urban settings is crucial because the results may guide the allocation of scarce resources toward future vector control.

The spatiotemporal patterns of dengue spread in Brazilian settings are limited and mostly based on serological prevalence and incidence data (*Barreto & Teixeira, 2008*; *Teixeira Mda et al., 2002*; *Teixeira et al., 2013*). Some recent studies address this topic in a larger scale (*Nunes et al., 2012*; *Nunes et al., 2014*). So far, only one work addresses the spatial dynamics of an urban dengue outbreak in the city of São Jose de Rio Petro using viral genetic data (*Mondini et al., 2009*). These studies are imperative because socio-demographic and ecological factors affect diffusion dynamics (*Cuong et al., 2013*; *Jeefoo, Tripathi & Souris, 2011*; *Raghwani et al., 2011*; *Rasmussen, Boni & Koelle, 2014*; *Schreiber et al., 2009*; *Vazquez-Prokopec et al., 2010*). In the present work we describe an outbreak of DENV-4 during 2013 in the city of Guarujá, Brazil, following Bayesian phylogenetic analysis of envelope gene sequences. Our results emphasize the importance of real-time follow up and guided actions to achieve better control during epidemics.

## METHODS

### Study site

Guarujá (23°59′37″S 46°15′23″W) is a coastal city in Santo Amaro Island, situated at the shore of the state of São Paulo, Brazil (Fig. 1A). The city is embedded in a tropical rain forest. It has a tropical humid climate that is characterized by having high average air temperature and rainfall. The average annual temperature is 24.7 °C (Min 18 °C | Max 31.3 °C) and the annual rainfall is 3,400 mm; February is the wettest month (average rainfall of 413 mm) and August is the driest one (average rainfall of 156 mm). The city main economic sources are seasonal tourism and port related activities. The estimated population in 2013 was 306,683 and the human population density was around 2,000 inhabitants per $km^2$. Official dengue figures by the Epidemiological Surveillance Center of
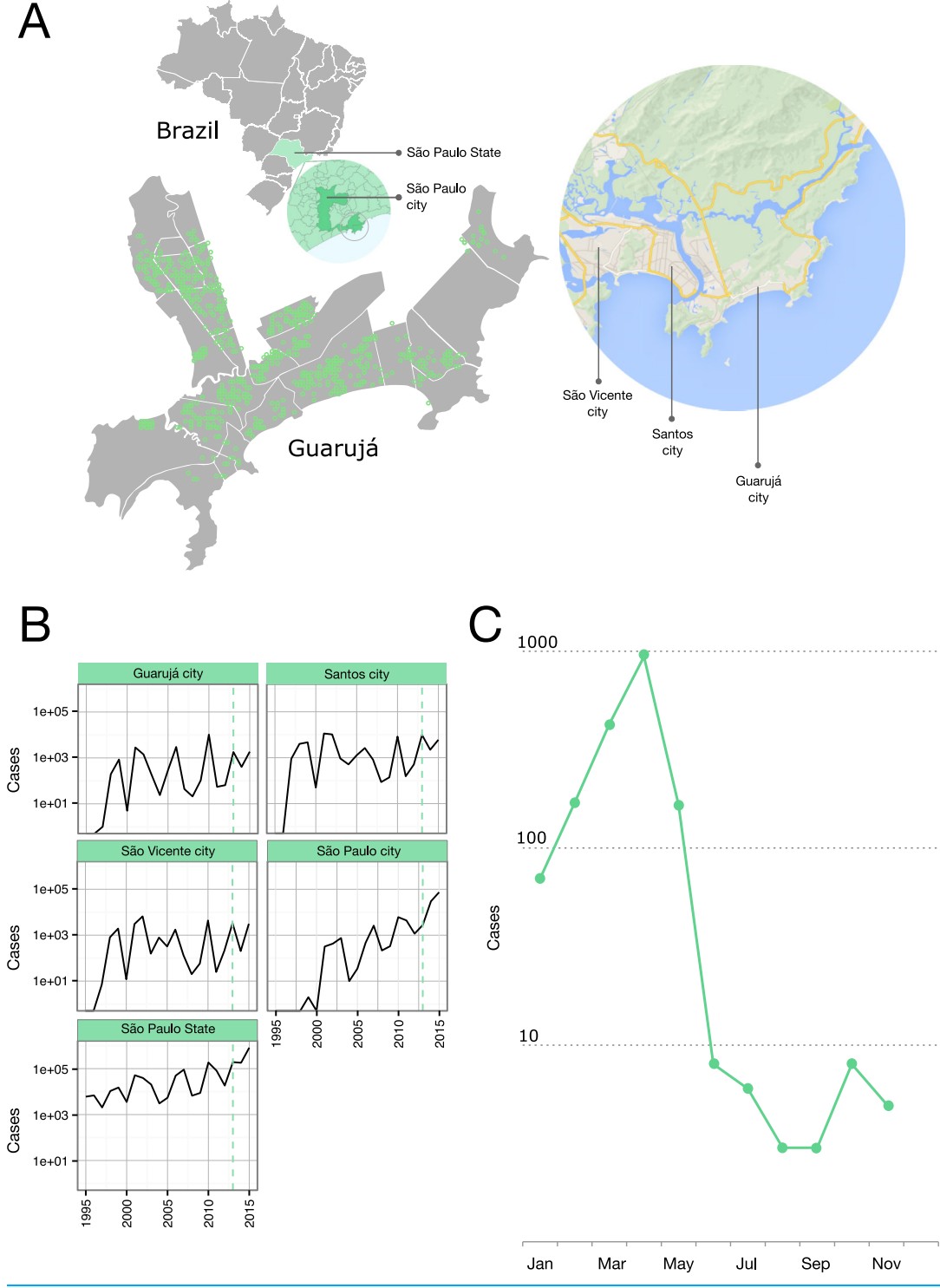

**Figure 1 Epidemiological situation in the Municipality of Guarujá.** (A) Map of Guarujá and nearby localities; georeferenced samples are plotted over the map (see File S1 for displaying in Google Earth). (B) Yearly dengue notifications for the State of São Paulo and some municipalities; dotted line intersects 2013. Fisher exact tests and corrections for multiple comparisons (Bonferroni and Benjamini-Hochberg) were used to assess the significance of the reduction in the number of cases from the previous epidemic year; tests were done by municipality (n = 429) and compared to the records for the State; Guarujá, São Vicente and São Paulo were among the municipalities (n = 103) with a significant reduction of notifications (p < 0.05). (C) Monthly dengue notifications for the municipality of Guarujá during 2013.

the State (CVE) date back to 1997 and sum 24,000 total cases up to 2012. There is no detailed information concerning the previous exposure to distinct DENV serotypes and the municipality relies on the surveillance by the Adolfo Lutz Institute (The Central Public Health laboratory from The State of São Paulo).

## Sample collection

In late 2012, our group at University of São Paulo joined efforts with the Guarujá Municipality's office of epidemiological surveillance and with a local clinical laboratory analysis center (Itapema) to map the incidence of dengue in the city and obtain viral genetic data. Both institutions contributed with the collection of samples citywide and by performing preliminary immunochromatographic diagnostic tests for dengue (*Kassim, 2011*).

Patients of any age with symptoms and signs of dengue disease that were examined in Primary Health or Emergency Care Units and tested positive for IgM and/or NS1 were considered for the study. Symptoms included fever, frontal or retro-orbital headache, severe pain (muscles, bones, legs, joints, lower back or abdominal), nausea, vomiting, taste disturbance and anorexia. Signs included high fever (usually between 38.5 and 41 °C) persisting for 24 h, rash, hemorrhagic manifestations, hypotension and narrow pulse pressure.

## Ethics statement

The Ethical Review Board of the Biomedical Science Institute at University of São Paulo approved this study (Statement 933/CEP). All adult subjects provided an informed written consent, and a parent or guardian of any child participant provided the written informed consent on their behalf.

## Molecular testing

Viral RNA was extracted from serum samples with the QIAmp viral RNA mini kit (Qiagen, Venlo, Limburg, Netherlands) and the complementary DNA was synthesized using the SuperScript® VILO™ cDNA Synthesis Kit (Life Technologies, Carlsbad, California, United States). The GoTaq® Green Master Mix (Promega, Madison, Wisconsin, United States) was used for PCR amplifications of the envelope gene using the primers of *Bennett et al. (2003)*. The ExoSAP-IT reagent was used for PCR Product Cleanup (Affymetrix, Santa Clara, California, United States) and the sequencing reaction was performed using the BigDye® Terminator v3.1 Cycle Sequencing Kit (Life Technologies). Sequencing reaction products were purified using the BigDye XTerminator Purification Kit (Life Technologies) and sequenced on an ABI PRISM® 3130 Genetic Analyzer (Life Technologies). Contigs were assembled using the program Codon Code aligner (www.codoncode.com/aligner).

## Genetic analysis

Sequences were aligned using Muscle 3.8.31 (*Edgar, 2004a*; *Edgar, 2004b*) followed by visual inspection and manual editing with Mesquite 2.75 (*Maddison & Maddison, 2014*). Polymorphisms were analyzed with DNASP 5 (*Librado & Rozas, 2009*). The Tajima D statistic test was used to evaluate deviations from the neutral expectation of

molecular evolution (*Tajima, 1989*). The package HyPhy v2.2 was used to screen for recombination (SBP-Single Breakpoint Recombination and GARD-Genetic Algorithms for Recombination Detection) and for positive selection; the dengue strain H780090 isolated in Boa Vista, RR-Brazil (29 November 2010) was used as a reference for selection analysis. Both genealogy-based, codon-site models Single Likelihood Ancestor Counting (SLAC) and the Fixed Effects Likelihood (FEL) methods were used to estimate the non-synonymous (dN) and synonymous (dS) rates of substitution (*Delport et al., 2010*; *Kosakovsky Pond & Frost, 2005*; *Pond, Frost & Muse, 2005*).

## Phylogenetic analysis

The sequences obtained in this study were combined with a DENV-4 database from a previous study (*Villabona-Arenas & Zanotto, 2011*) to identify the genotype. This was achieved using high-throughput clustering with the UCLUST algorithm in the package USEARCH (*Edgar, 2010*).

The JModeltest software was used for the statistical selection of the best-fit model of nucleotide substitution under the Akaike information criterion (*Darriba et al., 2012*; *Guindon & Gascuel, 2003*). Sequences were dated according to the day of sampling and used for phylogenetic reconstruction and the estimation of the rate of evolutionary change ($\mu$) (subs/site/year) using Bayesian Inference (IB) in Beast v2.3.1 (*Bouckaert et al., 2014*); the tree prior was a Birth-Death with Serial Skyline Sampling (BDSKY) (*Stadler et al., 2013*) (see Table S1 for parameterization). A Bayesian maximum clade credibility (MCC) tree was inferred from a set of plausible trees sampled at the stationary phase of four independent Markov Chain Monte Carlo (MCMC) runs with 200 million generations each using a relaxed (uncorrelated lognormal) molecular clock (*Drummond et al., 2006*). The convergence of parameters was assessed using Tracer v1.6 program (http://tree.bio.ed.ac.uk/software/tracer/) until all parameters estimates showed effective sample size (ESS) values over 200.

## Time-varying reproduction numbers

We used the approach of *Cori et al. (2013)* to estimate the instantaneous reproduction number using the R-package EpiEstim (*Cori et al., 2013*; *Salje et al., 2012*). The method requires incidence data and a serial interval distribution (a gamma distribution with shift 1) that describes the time between the onset of symptoms in a primary case and the onset of symptoms of secondary cases. We used the weekly notifications from the municipality's office of epidemiological surveillance for the epidemiological year of 2012–2013 (daily data was not available) and a serial interval distribution that reflected previous estimates of the dengue incubation period (Table S1); the outcome of a censored Bayesian time-to-event model estimated the dengue intrinsic incubation period (the time between a human being infected and the onset of symptoms due to the infection) around six days (95% CI 3–10 days) and the best-fitting temperature-dependent extrinsic incubation period around 6.5 days (95% CI 2–15 days) at 30 °C (*Chan & Johansson, 2012*). The analysis took into account the uncertainty in the serial interval distribution by integrating over a range of means and standard deviations of the serial interval (the mean and standard deviation were allowed to vary according to truncated normal distributions; Table S1)

(*Cori et al., 2013*). We used a gamma prior distribution (Table S1) for the reproduction number that includes previous estimates (between 1.33 and 11.6) (*Halstead, 2008*).

### Spatiotemporal dispersion pattern

In order to find the most parsimonious set of rates explaining the diffusion process along the sampled trees for a geolocated dataset, we used the Bayesian stochastic search variable selection (BSSVS) approach as implemented in Beast v2.3.1 (*Bouckaert et al., 2014*). The method assumes exchange rates in a continuous-time Markov chain (CTMC) to be zero with some prior probability (*Lemey et al., 2009*) and performs ancestral reconstruction on a single character, which represents the location of the taxa. Sampled taxa are associated with locations and the ancestral states of the internal nodes in the sampled trees can be reconstructed from the taxon locations; because the sampled trees are in units of time we can reconstruct the diffusion over time by following character transformation over the branches (*Drummond & Bouckaert, 2015*). Locations were represented by discrete groups of adjacent neighborhoods (discrete phylogeography) and a Bayes factor (BF) test was run to identify the rates contributing to the migration path with the software Spread v1.0.4 (*Bielejec et al., 2011*). The number of neighborhoods was reduced to a maximum of 10 localities (chosen by vicinity a by number of samples reported) in order to diminish sample-size bias (disproportionate sampling can strongly bias phylogeographic analyses because over sampled populations will more likely to be inferred as source populations).

## RESULTS

### Sampling

The year 2013 coincided with a steep rise in the confirmed cases of dengue fever in the State of São Paulo (Fig. 1B). Public Health authorities of Guarujá reported a total of 1,805 autochthonous dengue cases during this year.

We studied 505 PCR dengue-positive patients during the study In Guarujá. These samples were collected between December 2012 and July 2013. Serotyping determined 10 (1.9%) to be DENV-1, eight (1.5%) to be DENV-2, two to be DENV-3 (0.4%) and 505 to be DENV-4 (96.2%). DENV-4 was relatively new to the country and outbreaks had been reported throughout the country since it was first detected in Brazil in 2011. Preliminary results reporting the documentation of the co-circulation of the four serotypes was published elsewhere (*Villabona-Arenas et al., 2014*).

Complete envelope (E) gene sequences were obtained for 354 DENV-4 (1,485 bp-long). The remaining DENV-4 was not processed due to technical problems (*e.g.*, did not yield sufficient viral RNA). These sequences were deposited in GenBank under the accessions KP703864–KP704217.

### DENV-4 genetic diversity

A total of 109 sites (7.3%) of the envelope gene were polymorphic; 32 sites (2.1%) fell in the first and second codon position and 77 (5.2%) fell in the third codon position. There were a total of 95 haplotypes and 75 of them represented one unique sequence; the

three most frequent haplotypes had 119 (34%), 38 (11%) and 30 (8.5%) sequences (Haplotype diversity, the probability that two haplotypes drawn uniformly at random from the population are not the same, was 0.86). The Tajima's D value was of −2.56 (p < 0.001) (−2.48 for the combination of both first and second codon positions, −2.26 for the third codon position) evidenced an excess of low frequency polymorphisms relative to expectation, indicating population size expansions and/or purifying selection. In agreement with this, the overall rate of non-synonymous over synonymous changes $dN/dS$ value of 0.15 (95% CI 0.10–0.21) for the entire gene suggested purifying selection. A few codons, which fell in the central and dimerization gene domains, showed statistically significant purifying selection (codons 92, 133, 184 and 225) at the significant level of 0.05. Although a few sites experienced an elevation on $dN/dS$ there was no statistical evidence for adaptive evolution.

## Evolutionary history and epidemiological dynamics

The 354 DENV-4 local sequences fell into the Latin-American cluster of viruses of Genotype II together with other Brazilian samples (see Data S1).

The best-fit model of nucleotide substitution was Tamura-Nei (TrN) with invariables sites. The mean evolutionary rate was $2.79 \times 10^{-3}$ substitutions per site per year (95% HPDs: $2.06 \times 10^{-3}$–$3.77 \times 10^{-3}$ substitutions per site per year). The estimates for the epidemiological parameters were: the sampling proportion was 1.4% (95% HPDs: 0.03–3.1%), the infectious period was 6.7 days (95% HPDs: 5–10 days) and the origin of the epidemic was the 21 December 2012 (95% HPDs: 16 December 2012–26 December 2012). Figure S1 show the extent to which prior information matched the posterior. Sampling from the prior analysis indicated that the posterior and prior traces were the same and that the overall constraints were not forcing the results.

Figure 2A shows the MCC tree with two clades early on in the epidemic. The mean time to the most recent common ancestor of these clades did not differ significantly, suggesting that both viral lineages diverged over similar time-scales, and then co-circulated.

The dynamic of R is presented in Fig. 2B. A value of the parameter R over 1.0 indicates that the disease will be able to spread in a population. For the time series analysis, the estimates are high during the first four months of the year and the curve decrease rapidly by the end of April with values below 1.0 in May and June. We did not included the birth-death skyline plots because this reconstruction may be misleading around the time we had severe sampling issues (due to the switch to clinical diagnosis) and not enough phylogenetic diversity was observed within the sampled genetic data (*du Plessis & Stadler, 2015*); additional BDSKY analyses where sampling proportions were estimated in a piece-wise manner over six different intervals did not solve this. Moreover, these methods not always reconstruct complex dynamics when other factors such as seasonality, spatial structure and vector dynamics are not incorporated (*Rasmussen, Boni & Koelle, 2014*).

## Phylogeography of DENV-4 over the city

We were able to geolocate 286 patients (81%) (Fig. 1A) based on the addresses recorded by the Guarujá Municipal Health Department (Records were not available for the

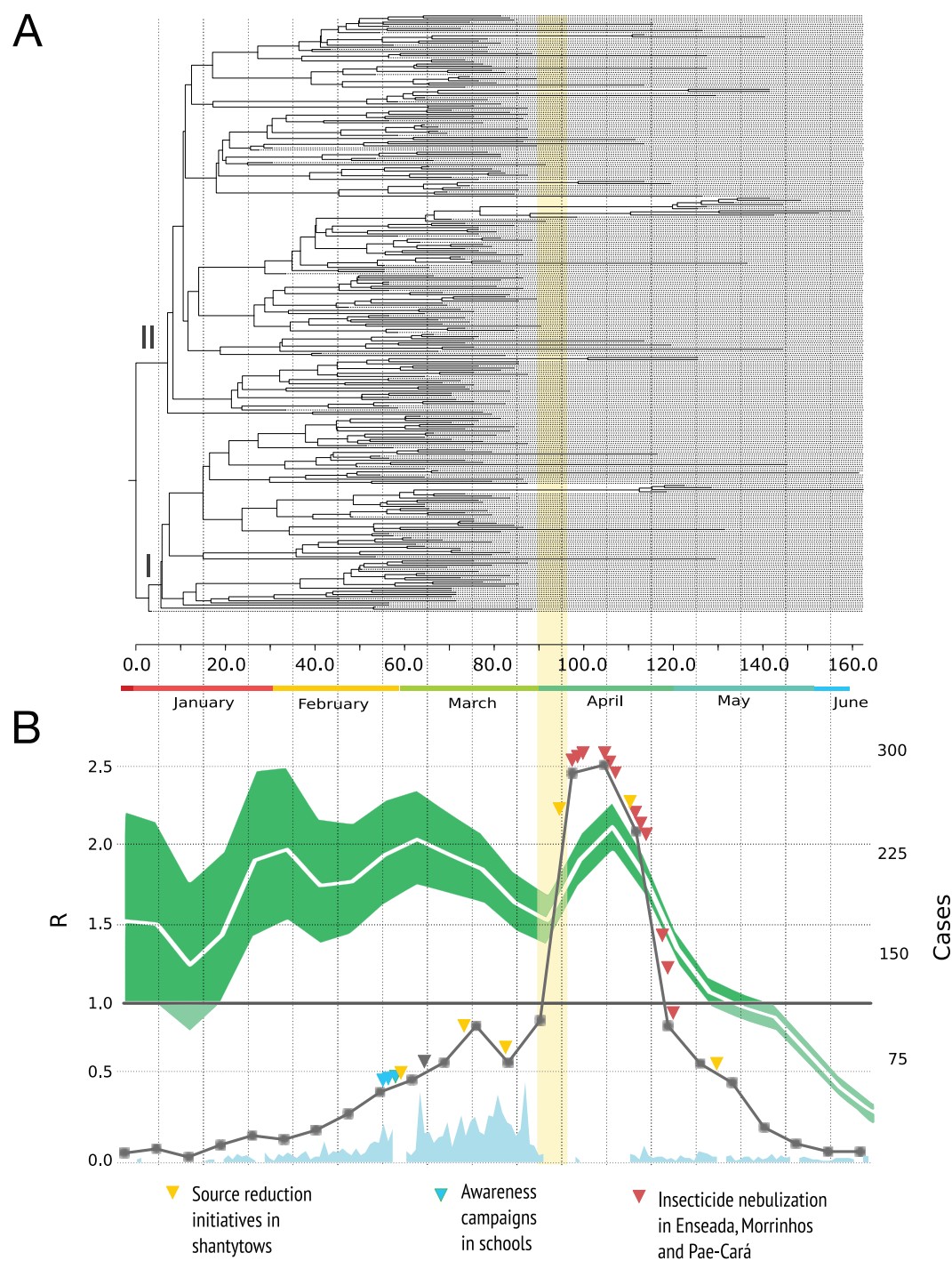

**Figure 2 Phylogenetic relationships and reproduction numbers of DENV-4 genotype II isolated in the municipality of Guarujá from January–June 2013.** (A) Maximum clade credibility (MCC) tree inferred using envelope gene sequences. Branch tips were removed for simplicity. (B) Median estimates and 95% IC for the effective reproductive number using incidence time series data. For (B) official dengue reports done by epidemiological week (gray squares) and sampling done in a daily basis (sky blue) are presented. The band (yellow) represents the period in which the epidemic alert was announced. The gray triangle informs when the neighboring city of Santos announced its own epidemic alert.

remaining patients); Fig. 2B compares the actual number of official cases reported and our DENV-4 geolocated sampling. Our first geolocated sample was collected in 2 January 2013 at the neighborhood Enseada. This location has a high number of residents (20,883 based on the 2010 census records) and is home to the largest beach concentrating mostly residents. Pae-Cará together with its neighbor Itapema, are the neighborhoods with more residents (26,054 and 26,070 respectively) followed by Morrinhos (24,387), Enseada and Jardim Boa Esperança (20,753). The digital map was provided by the Municipality's office and represents the master plan for development and urban planning in the city. Figure 3 illustrates the overall discrete spatial diffusion over the urban area. These results were gauged from a full location-annotated MCC tree, available as Fig. 4, which evidenced an early widespread distribution of the virus in January. The introduction events into each discrete unit are depicted in Fig. 3A; these represent viral diffusion during the onset of the epidemic. These figures suggest that two localities, Enseada and Pae-Cará, were key virus sources. Later on, all regions become interconnected in terms of viral diffusion. The adjacent high-income, low-population density neighborhoods (Jardim Acapulco and Pernambuco) had no cases sampled. This is not explained by distance or lack of connection, because low-income areas nearby (the shantytowns in Mar e Céu, to the South, and Pereque Beach, to the North) had several cases during the epidemic. Bayes factor test of significant diffusion rates shows that another two localities (Morrinhos and Jardim Boa Esperança) played an important role during the course of the epidemic (Fig. 3B). The initial diffusion pattern reproduced to some degree the main access highways of the island: a north-south axis with Pae-Cará and a west-east axis over the littoral with Enseada (Fig. 3C).

## DISCUSSION

In the present study we described the outbreak of DENV-4 during 2013 in the city of Guarujá, Brazil. During the outbreak we documented purifying selection and found no statistical evidence of adaptive evolution. Nonetheless, inferences about selection and rates drawn from the analyses should be interpreted with caution. The observed differences between our sequences may represent segregating sites in a population and under this scenario evolutionary rates can be overestimated and dN/dS ratios below one can be found under both negative and positive selection (*Kryazhimskiy & Plotkin, 2008*).

On April 4th, the Public health authorities of Guarujá announced the epidemic alert (when a city reaches the incidence of 100 cases per 100 thousand inhabitants) and during this month they strengthened control measures. Confirmatory diagnoses for dengue became clinical (ignoring the possibility of other acute febrile illnesses) and the number of cases increased dramatically after the epidemic alert was announced.

The reproduction number dynamics (Fig. 2B) and the timing of the coalescent events (concentrated around February) point to an epidemic that started much earlier than the case report records. A comparable observation was done in another Brazilian setting (*Mondini et al., 2009*): the epidemic peak by demographic skyline methods took place around two months before the epidemic peak by case report data. The authors

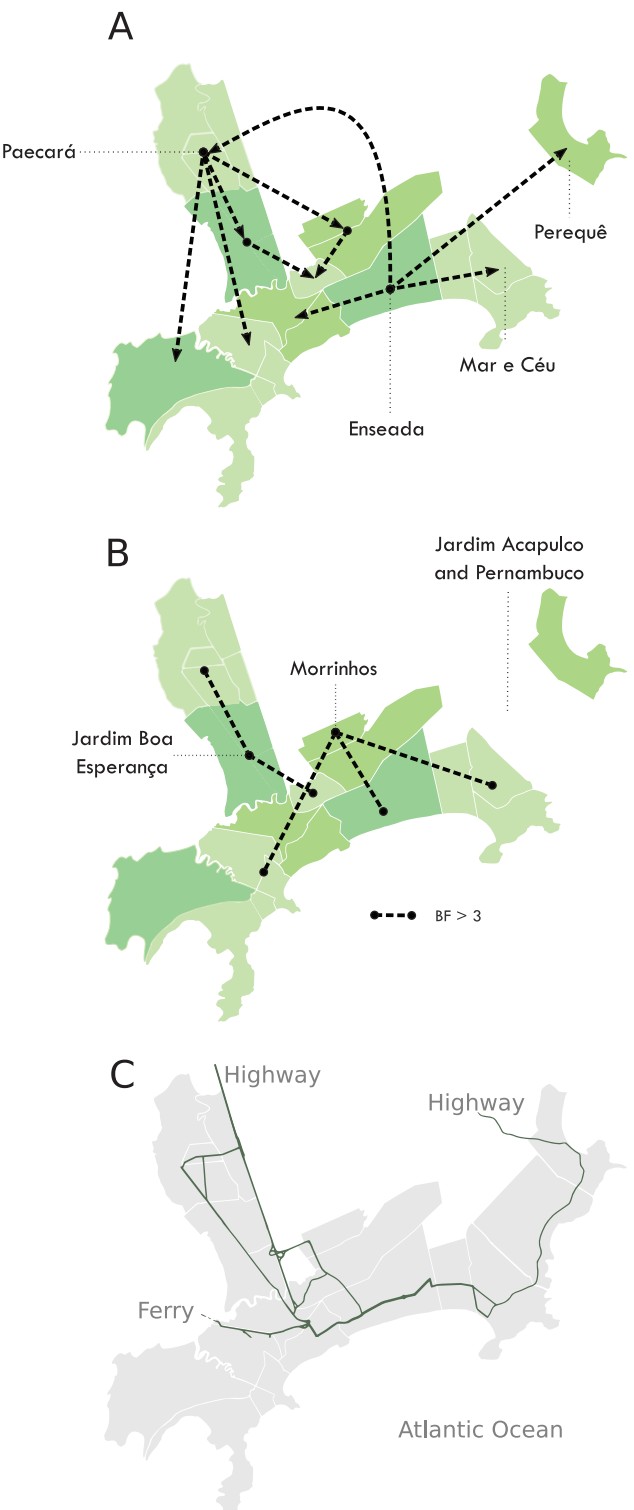

**Figure 3** **Diffusion of DENV-4 genotype II in the municipality of Guarujá from January–June 2013.** Discontinuous green areas represent discrete areas. (A) Introduction routes into each area. (B) Routes that best explain virus diffusion all over the city. The reconstruction was done following a location-annotated MCC tree available as Fig. 4. (C) The main avenues and highways of Guarujá. Names are given for the areas that are quoted in the text.

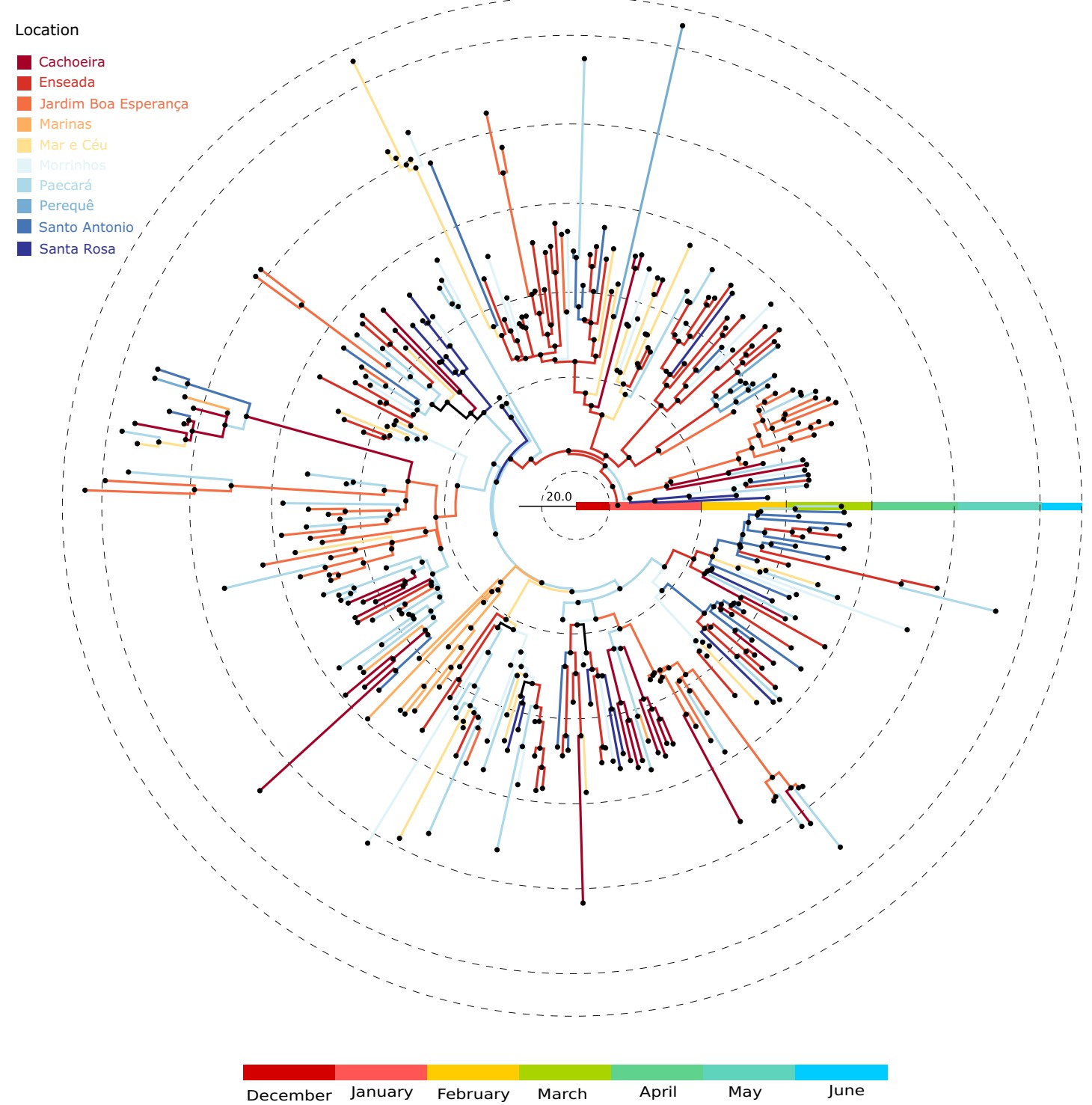

**Location**
- Cachoeira
- Enseada
- Jardim Boa Esperança
- Marinas
- Mar e Céu
- Morrinhos
- Paecará
- Perequê
- Santo Antonio
- Santa Rosa

December   January   February   March   April   May   June

**Figure 4 Location-annotated Maximum clade credibility tree.** Branch color corresponds to locations; tips were removed for simplicity.

argue that such finding resulted from an increase in false positives after the epidemic alert and we argue that in Guarujá the population awareness might have also contributed to a sudden increase.

Higher values in reproduction numbers (or relative genetic diversity estimates) preceding the peak of laboratory-confirmed cases may reflect virus population spread in a large unreported infected population. Dengue virus infection results in more asymptomatic cases than symptomatic ones and this difficult the early detection of increased incidence. This phenomenon has been documented for Brazilian urban settings (*Endy, 2002*; *Poblap et al., 2006*; *Teixeira Mda et al., 2002*). The spatial diffusion analysis shows that when a significant number of clinical cases began to appear, the virus was practically distributed throughout the city (Fig. 4). Moreover, by looking at the available data on dengue from 2012 (Fig. 1C) it is clear that transmission is sustained all over the year. Following the dynamics of the reproduction number using incidence time series, we observed that the disease spread quickly during most of the observational period and decreased by the end.

During November 2012, the city led the ranking of mosquito infestation in the State of São Paulo and the averaged Breteau index for *Aedes aegypti* during 2013 was high (3.12). Source reduction initiatives started in shantytowns during March but the flow of people and of mosquitoes from other areas may have offset their contribution. Municipality health authorities considered the use of massive insecticide nebulization when extensive symptomatic cases were recognized. The use of mosquito fogging trucks is not executed citywide because it is costly and relies on availability from the Adolfo Lutz Institute, therefore strategic areas have to be selected. The neighborhoods of Enseada, Paé-Cara and Morrinhos were selected because they reported higher numbers of reactive NS1 antigen results; four rounds of insecticide nebulization were applied during April and massive insecticide nebulization coincided with a rapid reduction in the number of cases over time. It may be argued that the outcome was due to the depletion of the susceptible population or only a change in seasonality (*Egger et al., 2008*). Nonetheless, Guarujá was among the municipalities (103 out of 429) that showed a statistically significant reduction of notification when comparing the figures of the previous epidemics (2010) and those of the State (a naïve-experienced population to DENV-4). Overall, this suggests that control strategy achieved some degree of control.

The research into their effectiveness of vector control is scarce and almost nothing is known about how well it reduces DENV transmission (*Achee et al., 2015*). The two high-income neighborhoods that had privately-owned and hired vector control services (as informed by the health authorities) showed a discontinuity when pinpointing the dengue incidence over the city. This suggests that control actions, when applied in a timely and sustained manner, are useful. Rapid and unplanned urbanization (*e.g.* shantytowns) has provided appropriate circumstances (high population density and high contact rates between humans and mosquitos) for substantial vector breeding in Guarujá and several municipalities of Brazil. For example, nearby cities, such as Santos and São Vicente (Fig. 1B), also have important number of dengue notifications throughout the year and therefore a continuous flux of infected people and mosquitoes is expected. Under this scenario, city public health administrations with limited resources encounter a big challenge. In Guarujá, local authorities prioritized vector control based on georeferenced viremic patients and the epidemic of an expected higher magnitude. To achieve full control however, this may be not enough, but it is a first coherent step.

## CONCLUSION

Studying urban outbreaks is important; successful public health interventions require detailed knowledge of the disease dynamics and how it spread within the population. It is very difficult to stop dengue spread because *Aedes* mosquitoes bounce back to initial numbers after control interventions and because unapparent infections also contribute to DENV persistent circulation (*Duong et al., 2015*). Nonetheless, in the absence of a vaccine, source reduction initiatives and massive control actions are the options that city public health administrations have. We have evidenced that a delayed response may result in an epidemic that grow beyond the capabilities of local health authorities but that sound efforts may diminish its effect.

## ACKNOWLEDGEMENTS

We thank all the participants in this study for generously helping us in this research. We also thank all the medical staff and administrative staff from the Primary Health Care Facilities, the Emergency Care Units and the ITAPEMA clinical laboratory analysis center who have kindly assisted us by collecting blood samples from participants. We thank the Guarujá Municipality's office of epidemiological surveillance for the discussions and the practical and technical advice.

### Funding

This work was supported by FAPESP (Fundação de Amparo à Pesquisa do Estado de São Paulo) grant 2010/19059-7. CJVA and JLO received the FAPESP scholarships 2011/17071-2 and 2013/10382-8 respectively; PMdAZ holds a CNPq–PQ scholarship. The funders had no role in study design, data collection and analysis, decision to publish, or preparation of the manuscript.

### Grant Disclosures

The following grant information was disclosed by the authors:
FAPESP (Fundação de Amparo à Pesquisa do Estado de São Paulo): 2010/19059-7.
CJVA: 2011/17071-2.
JLO: 2013/10382-8.

### Competing Interests

The authors declare that they have no competing interests.

### Author Contributions

- Christian Julián Villabona-Arenas conceived and designed the experiments, performed the experiments, analyzed the data, wrote the paper, prepared figures and/or tables, reviewed drafts of the paper.
- Jessica Luana de Oliveira performed the experiments.
- Carla de Sousa-Capra conceived and designed the experiments.

- Karime Balarini performed the experiments.
- Celso Ricardo Theoto Pereira da Fonseca conceived and designed the experiments, contributed reagents/materials/analysis tools.
- Paolo Marinho de Andrade Zanotto conceived and designed the experiments, analyzed the data, contributed reagents/materials/analysis tools, wrote the paper, reviewed drafts of the paper.

### Human Ethics

The following information was supplied relating to ethical approvals (i.e., approving body and any reference numbers):

The Ethical Review Board of the Biomedical Science Institute at University of São Paulo approved this study (Statement 933/CEP). All adult subjects provided an informed written consent, and a parent or guardian of any child participant provided the written informed consent on their behalf.

### DNA Deposition

The following information was supplied regarding the deposition of DNA sequences:

These sequences were deposited in GenBank under the accession numbers KP703864–KP704217.

### Supplemental Information

Supplemental information for this article can be found online at http://dx.doi.org/10.7717/peerj.1892#supplemental-information.

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
