# Peer review of "Epidemiological dynamics of an urban Dengue 4 outbreak in São Paulo, Brazil"

_PeerJ, doi:10.7717/peerj.1892_

## Round 0.1 · original submission · Major Revisions

Your manuscript has been critically evaluated by three independent reviewers. They had several critical comments that should be addressed in a revised version of the manuscript. In particular, all reviewers had concerns about the discrepancy between the estimated reproductive numbers using Birth-Death skyline methods and incidence time series data. Please take care in clarifying this aspect of your study and discussing potential factors that could have led to the discrepant finding.

Reviewer 1 ·

Basic reporting

It's fine

Experimental design

Methods are not described in sufficient detail. Please refer to full review for details.

Validity of the findings

I have serious doubts about the results. Please see full comments.

Additional comments

This was an interesting report of an outbreak in a small city in Brazil, in which the authors ran sophisticated bioinformatics analyses to understand the temporal and spatial dynamics of the outbreak. I felt that some aspects of the analysis were not justified and there are certainly some issues with their use of these methods that I discuss further below. Given my serious concerns about some of the analysis and findings, I cannot recommend publication until the authors describe their methods in enough detail to reproduce and assess the robustness of their inferences.

Major points
1. Elaborate on the entry criteria [L112—5]. Patients with any sign or symptom from this list were tested for dengue infection? Or just those the clinician suspected had dengue?
2. Estimates of the reproduction number. First, the methods are not described to enough detail for an expert to reproduce. E.g. they write that the serial distribution parameters were 2.0, 0.5, but do not define what these parameters are (nor do they define what distribution they are using in the first place). Nor do the authors describe how those parameters were derived. They appear suspiciously round. What data were used to obtain these numbers and how justified are they?
3. The 95% ‘CI’ (presumably credible interval) reported for the intrinsic incubation period is very wide (3 to 10 days). What exactly does this correspond to? If it is the mean incubation period it appears to be far too wide to be credible. I can only assume it is meant to describe quantiles of the distribution, but if that is the case, those quantiles themselves should have uncertainty intervals on them and the numbers need to be used with great care (i.e. as quantiles not as means). I worry the authors are conflating stochasticity with uncertainty. Similarly for the extrinsic incubation periods.
4. Similarly the authors have used a “prior distribution (2.0, 5.0)” [sic] for the reproduction number. Again, two numbers do not a distribution make. What family of distributions was used, what is the specific parameterization, and from what data were these two numbers obtained? The authors mention a paper by Halstead and a range of values for R (1.33 to 11.6). It is not clear how this range corresponds to the numbers 2.0 and 5.0.
5. The description of the parameters in the BDSKY model does not make sense. The ‘parameterisation consisted of 3 correlated parameters that can have a different number of changes’. Firstly, parameters are not correlated, they are just parameters. If they have a distribution, their joint distribution can be correlated. If that is what the authors mean, that should be made explicit. Secondly, what do they mean by the parameters can ‘have a different number of changes’? The names of the parameters appear unusual and unclear: the ‘become uninfectious rate’ would usually be called the recovery rate, while the ‘sampling proportion’ should be made much more explicit: what are the numerators and denominators? For all parameters, informative priors have been set, but there is no justification provided for the particular forms used. As the priors appear to influence the posteriors, it behoves the authors to provide very concrete justifications for any numbers used, and if any are not data driven, to make those ‘made up’ parameters very clear to the readers so that we can ignore them in interpreting the results.
6. Similarly the authors report that they used ‘one dimension’ for two parameters and ‘six’ for another. If the parameter R is a scalar, it only has one dimension and so for them to ‘use’ six is completely unclear. I can only imagine what they have done (perhaps they set R_t to be time-varying and a function of t and six other parameters, or perhaps they used a proposal distribution that was a mixture of six kernels?). Readers should not have to use their imagination in reading scientific papers!
7. The effective sample size that they reach (200) is extremely small for robust inference to be conducted. What are the (Monte Carlo driven) standard errors on their estimates? Presumably they are intolerably wide.
8. The abbreviation CTMC is used (L209) in the methods section without being spelled out. It is unclear how the authors expected the reviewers to be able to review their manuscript properly without providing sufficient information on what they have done.
9. The rationale for reducing the number of neighbourhoods to reduce sample-size bias (L215) is not clear.
10. The estimates of s (L262) appear much too low. If I have understood them correctly, s is the proportion of cases that are reported. The best guess is 0.014 or about one in 70 infections. Based on the number of cases reported, that corresponds to about 40% of the population of the town infected over a 6mo period! Is that plausible? It seems incredible to me.
11. The ‘coupled people/mosquito infectious period’ of 6.7d is unrealistically low if it means, as I assume, the mean infectious period in humans plus the mean infectious period in mosquitoes. The infectious period is probably around 7d in people, including one or two days prior to symptom onset, and while no one knows how long mosquitoes are infectious out in the field, I’m sure mosquitoes are infectious for more than a day. This to me suggests a problem with either the priors the authors are assuming, or with the model formulation itself. The justification (L265) for using an informative prior distribution is not a valid reason (it relates to the structure of the model, not to the choice of prior distributions).
12. The description of figure S3 in the results (L267) is very strange. Thankfully, the priors do not ‘match’ the posteriors, as the authors write (which would mean there is little information in their data set so all their inferences are based on their made up prior distributions). So they need to think again about how they describe this figure. But the figure is still worrying as it suggests that for at least some of the parameters, the prior distribution may have a strong influence on the posterior (especially the two rates). If this is the case, and the priors are driven by a dataset from some other study, then this is completely fine. However, if they are just ‘made up’ (as they appear to be) then this means that the results of the study come from the authors’ imagination rather than solely from scientific data, and this I find very worrying. If the analysis is meant to be for the authors alone, then this is fine, but otherwise, readers have the reasonable expectation that the information presented is based on data from the outbreak.
13. I struggle very badly to see how the realtime estimates of the reproduction number are feasible. The first estimates presented show that in mid-March, R goes below 1, and yet the epidemic continues to grow for a further month. I cannot see how this is possible unless the authors are using an extremely unusual definition of R. I can see that with dengue, it’s a bit unusual due to the presence of two hosts, and maybe the entire infection cycle is being associated with the infection of the first host, which would have the effect of making the transmission chain appear to be offset by some period. But I can’t see this period being one month, and even if it were, I think the analysis is completely misleading as the control efforts focus mostly on the mosquito part of the transmission cycle. Can the authors clarify what is going on here?
14. Further evidence of how weird this is can be seen by the completely non-overlapping estimates of the realtime reproduction number (figure 1B, 1C). These are supposed to be estimates of the same thing, and yet the uncertainty intervals do not overlap for much/most of the epidemic. This tells us straight away that at least one of the two estimates is *completely wrong*. (And of course, it could possibly be both.) I can’t recommend publication with this kind of major inconsistency, which certainly makes me worried about the strong statements in the discussion about peak in risk being earlier than the peak in cases.
15. Details need to be provided on how exactly the diffusion patterns were generated (fig 2).
16. Some of the supplementary figures are nice and should appear in the main manuscript (given PJ is an online journal I see no reason to move them to an online appendix).

Minor points
There are quite a few language issues that should be addressed. These do not really affect the meaning but it would be better to have them proofread, e.g.
L53 ‘Following cartographic approaches…’ doesn’t make sense. Minor rephrasing needed.
L57 ‘populations’ --> ‘population sizes’
L58 Move ‘not only’ to after ‘mosquitoes’
L59 ‘the mosquitoes lay their eggs’
etc

L65 Add citation for the direct quotation.
L96 Round figures? I don’t think we need the .80 inhabitants per km2.
L101 Feedback for what?
L150 Cite Tajima?
L261/2 Units for rates
L285 ‘largest beach concentrating bathers’ is an amazing image but presumably not what the authors mean.
L356 ‘Unfortunately’ strange?
L369 There is literature on the bottle neck that might be cited.

·

Basic reporting

No comments

Experimental design

No comments

Validity of the findings

No comments

Additional comments

Major comment:

It seems very plausible that the sharp decline in R inferred during the middle of the epidemic using BDSKY may be due to a lack of sampling during this period. What if the sampling fraction s was allowed to vary in a piece-wise manner like R to take into account changing sampling fractions over the epidemic? This may lead to greater uncertainty in the estimates of R from the phylogeny, but it may help eliminate a false signal of epidemic decline.

Minor comments:

For the priors used in the time series analysis, parameters are given but not the actual distributions (i.e. Gamma, Normal, Uniform). These, along with their parameters, could be easily displayed in a Table for easy reference. Also please make sure the unit of time is apparent for rate/duration parameters.

Line 199-201. The sentence “We used one dimension for parameters s and y and six dimensions for parameter R” will be confusing to non-BEAST users. Perhaps better to say that R was estimated in a piece-wise manner over six different intervals, since we wouldn’t expect s and y to have more than one dimension unless they are also estimated in a piece-wise manner.

Line 233. There does not appear to be a Figure 1D in Figure 1.

Line 245. I would change this to “population size expansions and/or purifying selection”.

Line 248. “purifying selection” -> Or do you mean positive selection if most are under purifying selection anyways?

Line 259-268. In the Results, I would restate what each of the parameters is so the reader doesn’t have to look back and see what s and y stand for.

Line 406. “The effectiveness of vector control approaches is scarce…” -> Maybe better to say that research into their effectiveness is scarce.

Additional minor grammatical points

Line 55. “manifest apparently” -> Maybe better to say ‘are symptomatic’ or mayble ‘clinically apparent’?

Line 59. “the mosquito lays their eggs” -> the mosquitos lay their eggs

Line 65. Why is “1,468,873” in quotes?

Line 76. “only one work address…” -> only one work addresses

Line 323. “alter” -> alert

Reviewer 3 ·

Basic reporting

No Comments

Experimental design

No Comments

Validity of the findings

No Comments

Additional comments

This manuscript investigates spatial-temporal dynamics of a dengue 4 outbreak in San Paulo, Brazil, using both case report data and sequencing data. By estimating the key epidemiological parameter, i.e. the reproductive number R, and the spatial spread pattern, the authors conclude that a substantial proportion of cases are unreported and this lead to a late recognition of an outbreak. The authors further suggest that vector control on specific clustered geographic regions may help reduce epidemic impact.

This manuscript presents novel and interesting data which shed lights on the spatial and geographical spread of dengue in an urban setting. These data are potentially useful in terms of understanding the spread of this important tropical and subtropical disease, and making appropriate control measures. The data are rigorous analyzed, and the manuscript is clearly written (although some improvements shall be made; see below). Please see below for a couple of my concerns.

1. It strikes me that the estimated temporal changes in the values of R from sequence data is in stark contrast with the values estimated from case count data. Is there any reason/hypothesis could explain the pattern observed? The authors mentioned in discussion about the possibilities of sampling bias, lack of mutation during the period, the accuracy of the timing of the collection. However, I am not sure how these factors would play out to give such a contrasting pattern. Many of these factors would mostly affect the confidence interval estimated, e.g. lack of mutation would increase the estimated confidence interval, rather than completely reverting the temporal pattern of change. One possible explanation would be positive selection, which is known to reduce the estimated effective population size in a neutral model. But I don’t know if there’s any evidence (by looking at the tree shape for example) showing a certain lineage is positively selected. Anyway, it’s a very interesting/important question to figure out. I would encourage the authors perform some sensitivity analysis on the hypotheses postulated in the discussion to see any of them would lead to this pattern. And a thorough discussion on the differences between the estimation of R using two methods will be interesting/helpful for the reader to interpret the results.

2. A couple of presentation issues:
a. the method on how R is calculated from case count data is not very clearly presented. Some more details are needed for the reader to understand how goes into the calculation.
b. the results form the estimation of epidemiological parameters (e.g. R) from BEAST is not well presented. I would suggest to add a few suppl. Figures to show how the distributions (of those key parameters) look like after the Baysian MCMC run, to help the reader to understand the uncertainties in the values of estimated parame

---

## Round 0.2 · accepted · Accept

I would like to thank you for your careful revision that addresses all points raised by the three reviewers.